# Evaluation of the Toxicity Potential of the Metabolites of Di-Isononyl Phthalate and of Their Interactions with Members of Family 1 of Sulfotransferases—A Computational Study

**DOI:** 10.3390/molecules28186748

**Published:** 2023-09-21

**Authors:** Silvana Ceauranu, Alecu Ciorsac, Vasile Ostafe, Adriana Isvoran

**Affiliations:** 1Department of Biology Chemistry, West University of Timisoara, 16 Pestalozzi, 300115 Timisoara, Romania; silvana.ceauranu96@e-uvt.ro (S.C.); vasile.ostafe@e-uvt.ro (V.O.); 2Advanced Environmental Research Laboratories, West University of Timisoara, 4 Oituz, 300086 Timisoara, Romania; 3Department of Physical Education and Sport, University Politehnica Timisoara, 2. Piata Victoriei, 300006 Timisoara, Romania; alecu.ciorsac@upt.ro

**Keywords:** metabolism, molecular docking, di-isononyl phthalate monoesters, toxicological effects

## Abstract

Di-isononyl phthalates are chemicals that are widely used as plasticizers. Humans are extensively exposed to these compounds by dietary intake, through inhalation and skin absorption. Sulfotransferases (SULTs) are enzymes responsible for the detoxification and elimination of numerous endogenous and exogenous molecules from the body. Consequently, SULTs are involved in regulating the biological activity of various hormones and neurotransmitters. The present study considers a computational approach to predict the toxicological potential of the metabolites of di-isononyl phthalate. Furthermore, molecular docking was considered to evaluate the inhibitory potential of these metabolites against the members of family 1 of SULTs. The metabolites of di-isononyl phthalate reveal a potency to cause liver damage and to inhibit receptors activated by peroxisome proliferators. These metabolites are also usually able to inhibit the activity of the members of family 1 of SULTs, except for SULT1A3 and SULT1B1. The outcomes of this study are important for an enhanced understanding of the risk of human exposure to di-isononyl phthalates.

## 1. Introduction

Xenobiotics are chemical compounds that do not occur naturally in the human body and, if not metabolized, they could reach toxic concentrations. Humans are exposed daily to numerous xenobiotics: drugs, food additives, cosmetic ingredients, pesticides, environmental pollutants, etc. The metabolism of xenobiotics usually occurs through enzymatic reactions, and the enzymes involved are classified into phase I, phase II, and transporter enzymes. Phase I enzymes (such as cytochromes P450) convert lipophilic xenobiotics into more polar compounds for easier excretion and provide sites for conjugation reactions. Phase II enzymes (such as glutathione transferases and sulfotransferases) carry out the conjugation reactions and can interact directly with xenobiotics, but most of the time they interact with the metabolites obtained in the phase I metabolism [1]. 

This study focuses on family 1 of sulfotransferases (SULT1), containing enzymes that are known to catalyze the sulfate conjugation of pharmacologically important endobiotics and xenobiotics by transferring a sulfate group from the donor 3′-phosphoadenosine 5′-phosphosulfate (PAPS) to the hydroxyl group of an acceptor [2]. In humans, there are four subfamilies of SULT1: SULT1A (SULT1A1, SULT1A2, SULT1A3), SULT1B1, SULT1C (SULT1C1/C2, SULT1C3, SULT1C4) and SULT1E1 [3]. SULT1A1 and SULT1A2 have specificity for small phenolic compounds and SULT1A3 preferably sulfonates monoamines. SULT1B1 and SULT1C1 are involved in the sulfonation of thyroid hormones, whereas SULT1E1 sulfonates steroids, preferentially estrogens. 

Several polymorphisms have been identified for the SULT1 family of enzymes that proved to alter enzyme activities and presented differences in the metabolism of endogenous compounds, drugs and other xenobiotics [4,5]. The frequently identified SULT1A1 polymorphic variants are SULT1A1*1 (the wild type, WT), SULT1A1*2 (amino acids substitution R213H) and SULT1A1*3 (amino acids substitution M223V) [4]. Significant differences in the sulfation activities of SULT1A1*1, SULT1A1*2 and SULT1A1*3 have been identified, with SULT1A1*2 revealing lower catalytic activity and thermostability [5]. 

Among other xenobiotics, humans are exposed to phthalates because these compounds are widely used as solvents, additives and/or plasticizers in numerous consumer products. People can be exposed to phthalates by ingesting them from food or drink, by inhaling phthalate-contaminated air and dust resulting from paintings or furniture, by using pharmaceuticals and cosmetics, and by skin contact with products and/or clothing that contain them [6,7]. There are specific studies confirming the contamination of humans with phthalates as they reveal the presence of phthalate metabolites in human urine, saliva, breast milk and some serum samples [8,9,10]. Frequently observed harmful effects of phthalates on humans concern the reproductive, neurological and developmental systems [11]. A computational study conducted on 25 of the most used phthalates emphasized that they have the potential to interact with numerous molecular targets in the human organism (cytochromes, transporters, transcription factors, membrane receptors, kinases, phosphatases) [12]. These interactions may lead to harmful effects: skin and eye irritation, endocrine disruption, non-genotoxic carcinogenicity and toxicity of the respiratory and gastrointestinal tracts [12]. 

In humans, the metabolism of phthalates occurs in at least two steps: (i) a phase I hydrolysis when the diester phthalate is hydrolyzed into the monoester phthalate and (ii) a phase II conjugation [13]. Some in vitro and in vivo animal studies exposed that the monoester phthalates resulting from the metabolism of the diester phthalates are often more bioactive compounds [14]. Literature data reveal that benzylbutyl phthalate in high concentrations inhibits SULT1A1, whereas dibutyl phthalate and benzylbutyl phthalate inhibit SULT1E1 [15]. Furthermore, numerous phthalate metabolites revealed strong inhibition potential towards members of SULT1, especially SULT1A1, SULT1B1 and SULT1E1 [16]. SULT1A1 activity was strongly inhibited by mono-hexyl phthalate, monobenzyl phthalate, mono-octyl phthalate and mono-ethylhexyl phthalate. Mono-hexyl phthalate, mono-ethylhexyl phthalate, mono-octyl phthalate, mono-butyl phthalate and mono-cyclohexyl phthalate inhibited the activity of SULT1B1. Mono-hexyl phthalate, mono-ethylhexyl phthalate and mono-octyl phthalate inhibited the activity of SULT1E1 [16].

Di-isononyl phthalates (DiNP) are a group of isomeric forms of dinonyl phthalates widely used as plasticizers and found in various products such as cables, foil, PVC flooring, toys, shoe materials, tablecloths, etc. [17]. It was already assessed that upon ingestion, at least 50% of DiNP is expected to be absorbed through the gastrointestinal tract, with absorption decreasing as the dose increases (https://echa.europa.eu/documents/10162/31b4067e-de40-4044-93e8-9c9ff1960715, accessed on 15 June 2023). Investigations conducted on rats and on human adult volunteers have shown that DiNP has been metabolized to a number of secondary metabolites before being excreted in the urine: monoisononyl phthalate (MiNP), mono-hydroxy-isononyl phthalate (MHiNP), mono-carboxy-isooctyl phthalate (MCiOP) and mono-oxo-isononyl phthalate (MOiNP) [18,19,20]. 

This study aims to predict the possible toxicological effects on humans of the metabolites of DiNP and to assess the interactions of these xenobiotics with members of the SULT1 family, also taking into account the most frequently identified polymorphic variants of SULT1A1 enzymes. To obtain this information, a computational study was implemented.

Sometimes the reliability of the results obtained by using calculation methods is called into question. Considering both the cost of laboratory studies and the ethical concerns of using animals for testing, the role of computational studies is becoming well recognized. There has been an explosive growth in both the scale and diversity of data in the natural sciences, leading to the creation of freely accessible databases and enabling scientists to build accurate computational models for toxicology assessments. Consequently, the Organization for Economic Development and Cooperation (OECD) created guidelines for quantitative structure activity relationship (QSAR) models in 2004. The principles for building QSAR models and calculation methods have been described in detail since 2007 [21]. The Registration, Evaluation and Authorization of Chemicals (REACH) regulation has recognized QSAR techniques for studying the toxicological profile of chemicals [22]. Last but not least, molecular modeling methods offer the possibility to assess the risk to human health due to exposure to different chemicals as they can simulate possible modes of action. All these virtual screening tools have been useful for prioritizing bioassay requirements, saving resources and limiting the use of laboratory animals for testing. In this particular study, the computational approach was considered for a better understanding of the risk of human exposure to phthalates. 

## 2. Results

### 2.1. Physicochemical and Structural Properties of the Metabolites of Di-Isononyl Phthalate and of the Ligands That Are Present in the Crystallographic Structures of SULT1 Enzymes

The physicochemical and structural properties of the DiNP metabolites and of the ligands that are present in the crystallographic structures of SULT1 enzymes are presented in Table 1. 

Figure 1 reveals the molecular lipophilicity potential (MLP) on the molecular surface for the DiNP metabolites and for the ligands that are present in the crystallographic structures of SULT1 enzymes. In this figure, the hydrophobic parts of the surface are colored by violet and blue and the hydrophilic ones by orange and red.

Data presented in Table 1 and Figure 1 reveal that DiNP metabolites have physicochemical and structural properties that better resemble those of the 3,3′,5,5′-tetrachloro-4,4′-biphenyldiol, the SULT1E1 inhibitor. DiNP metabolites poses a higher number of rotatable bonds and consequently reveals a higher flexibility compared to that of 3,3′,5,5′-tetrachloro-4,4′-biphenyldiol. DiNP metabolites are bigger than dopamine, resveratrol and p-nitrophenol, which are the ligands that are present in the structural complexes of some of the SULT1 enzymes. It is important to notice that there are two molecules of p-nitrophenol in the structural file of the complex of SULT1A1*2 with this substrate. 

### 2.2. Toxicological Effects of the Di-Isononyl Phthalate and of Its Metabolites

Considering the fact that DiNP metabolites were observed in urine from human subjects, predictions about their toxicity on humans have been also obtained by using the ADMETLab2.0 computational tool [23,24]. This tool uses quantitative structure activity relationship regression or classification models for predicting the absorption, distribution, metabolism, excretion and toxicity (ADMET) profile of chemicals. For the ADMET properties that are predicted by the regression models, concrete predictive values are delivered, whereas for ADMET properties predicted by the classification models, the prediction probability values are provided. A positive value for the probability reveals that the studied molecule is more likely to produce the biological activity under investigation. Predicted probabilities with values higher than 0.700 are considered as reliable predictions [24]. 

The predicted toxicological effects of DiNP and of its metabolites on the human organism are illustrated in Table 2. In this table, there are delivered predicted concrete values for clearance (CL) and plasma protein binding (PPB) using regression models. Based on classification models, predicted probabilities are also computed for the ability of DiNP metabolites to penetrate the blood–brain barrier (BBB), to be inhibitors or substrates of P-glycoprotein (PgpI/PgpS), to produce cardiotoxicity by inhibiting the h-ERG channel (hERG), to produce drug liver injuries (DILI), to produce mutagenicity (Ames mutagenicity) and carcinogenicity, respectively, and to affect the nuclear receptors (NR). In the case of nuclear receptors, the probabilities that DiNP metabolites bind to the active sites of the androgen receptor (NR-AR-LBD), the aryl hydrocarbon receptor (NR-AhR), the estrogen receptor (NR-ER-LBD), the receptors activated by peroxisome proliferators (NR-PPAR-gamma) and, respectively, the probability to produce endocrine disruption (NR-aromatase) are predicted. 

The following can be observed from Table 2: (i) MiNP and MCiOP show a low elimination rate from human organisms as the computed values for clearance are low, 3.032 mL/min for MiNP and 1.369 mL/min for MCiOP; (ii) DiNp may be an inhibitor of P-glycoprotein as the computed probability for this effect is very high, 0.974; (iii) DiNP and its metabolites may strongly bind to plasma proteins, with all predicted values for binding being higher than 90%; (iv) all DiNP metabolites reveal the capacity to cause liver damage (the computed probabilities for producing this effect being higher than 0.700), with MiNP enlightening the higher probability to produce liver injuries; (v) all metabolites of DiNP reveal the potential to inhibit the receptors activated by peroxisome proliferators (computed probabilities to produce these effects are higher than 0.700); (vi) none of the listed metabolites show cardiotoxicity, mutagenicity, carcinogenicity, do not inhibit androgenic, estrogenic, endocrine and aryl hydrocarbon receptors, with the predicted probabilities corresponding to these toxicological effects being low.

### 2.3. Evaluation of the Interactions of DiNP Metabolites with the Family 1 of Human Sulfotransferases

The outcomes of the molecular docking are revealed in Figure 2, Figure 3, Figure 4, Figure 5, Figure 6 and Figure 7 and in Table 3. MCiOP and MiNP are able to bind to the active site of SULT1A1*1 (Figure 2), whereas MHiNP and MOiNP are not able to bind to the active site of this enzyme (Appendix A). 

In the case of the allelic variant SULT1A1*2, MiNP and MOiNP are the DiNP metabolites that are able to bind to the active site of the enzyme (Figure 3).

As a control, the molecular docking of p-nitrophenol (the substrate that is present in the crystallographic structure) to SULT1A1*2 emphasizes that the binding mode of p-nitrophenol obtained through molecular docking strongly corresponds to the position of one of the p-nitrophenol molecules in the crystallographic structure, the binding energy being −6.05 kcal/mol (Appendix A). The binding modes of the other two metabolites, MCiOP and MHiNP, that do not correspond to the active site of SULT1A1*2 are revealed in Appendix A. 

All the DiNP metabolites are able to bind to the active site of SULT1A1*3 (Figure 4).

In the case of SULT1A2, MCiOP, MiNP and MOiNP are able to bind to the active site of SULT1A2 (Figure 5). MHiNP does not bind to the active site of this enzyme (Appendix A). MHiNP has the higher topological polar surface area compared to the other DiNP metabolites; this property may be responsible for the fact that it does not bind to the active site of SULT1A2.

None of the DiNP metabolites bind to SULT1A3 and SULT1B1 active sites (Appendix A).

In order to assess the binding of DiNP metabolites to SULT1C1, first the molecular docking of iodothyronine (the typical substrate of SULT1C1) to the enzyme has been completed. The binding to the active site of SULT1C1 has been emphasized by superposition of the binding modes of iodothyronine to SULT1C1 with the structures of SULT1A1*2 and SULT1B1 (Appendix A). Further, the binding modes of DiNP metabolites to SULT1C1 were compared to the position of iodothyronine. 

All the DiNP metabolites are able to bind to the active site of SULT1C1 (Figure 6) and the binding poses correspond to the position of iodothyronine. 

All the DiNP metabolites are also able to bind to the active site of SULT1E1 (Figure 7), with the binding poses matching the binding position of the inhibitor 3,3′,5,5′-tetrachloro-4,4′-biphenyldiol.

The energies resulting from the molecular docking study and corresponding to the binding modes matching the active sites of SULT1 enzymes are presented in Table 3. 

Data presented in Table 3 emphasize the inhibition of SULT1A1*1, SULT1A1*2, SULT1A1*3, SULT1A2, SULT1C1 and SULT1E1 by the DiNP metabolites. Furthermore, MiNP is the metabolite that is able to inhibit almost all SULT1 enzymes, except SULT1A3 and SULT1B1, with the highest binding energy corresponding to SULT1A1*3. 

The results obtained using PLIP 2021 software regarding the identification of the residues of the SULT1 enzymes that are involved in the noncovalent interactions with DiNP metabolites are presented in Table 4.

Information presented in Table 4 confirms the good correlation between the complexes of SULT1 enzymes with their substrates resulted from docking and the complexes that are present in the crystallographic structures. 

## 3. Discussion

### 3.1. Toxicological Effects of the Di-Isononyl Phthalate and of Its Metabolites

To the best of our knowledge, this is the first study dealing with the possible toxicity of DiNP and its metabolites. Predictions obtained using the ADMETLab2.0 tool revealed that DiNp may be an inhibitor of P-glycoprotein, underlining that DiNP may potentially affect the absorption, distribution, metabolism and elimination of the substrates of this protein. The high degree of binding to plasma proteins of the DiNP and its metabolites may lead to long acting xenobiotics as their bound fractions are not available for metabolism or excretion. 

The present study emphasizes that the DiNP metabolites may produce liver damage and this outcome is in good correlation with literature data revealing that several phthalate metabolites produced liver injuries [25]. The potential of the DiNP metabolites to inhibit the receptors activated by peroxisome proliferators predicted in this study is also in good agreement with published data revealing the interactions of several monoester phthalates (including MiNP) with receptors activated by peroxisome proliferators [26]. 

None of the DiNP metabolites are considered to produce cardiotoxicity, mutagenicity or carcinogenicity. These predictions correlate with no carcinogenic [27] and mutagenic [28] effects noticed for some other phthalate metabolites. This study reveals no effects produced by the DiNP metabolites against the androgenic, estrogenic, endocrine and aryl hydrocarbon receptors. This finding is also in good correlation with specific literature data revealing no estrogenic and androgenic activities of mono-(2-ethylhexyl) phthalate [29].

Several toxicological effects on humans have been predicted for numerous other xenobiotics [30,31,32,33] and all these results should be used to guide the decision making regarding their use if human contamination is expected. 

### 3.2. Evaluation of the Interactions of DiNP Metabolites with the Family 1 of Human Sulfotransferases

A molecular docking study reveals that MCiOP and MiNP bind to the active site of SULT1A1*1, but MHiNP and MOiNP do not bind to the active site of this enzyme. This distinct behavior of MCiOP and MiNP that binds to SULT1A1*1 may be associated with their higher molecular lipophilicity potential (as presented in Figure 1); it is known that the distribution of lipophilicity in the molecular surface is important for the biological activity of a chemical compound. Literature data reveal that other monoester phthalates are strong inhibitors of SULT1A1*1: mono-hexyl phthalate, monobenzyl phthalate, mono-octyl phthalate and mono-ethylhexyl phthalate [16].

MiNP and MOiNP are the only DiNP metabolites that bind to the active site of the allelic variant SULT1A1*2. The binding energies for MiNP and MOiNP to the active site of SULT1A1*2 are higher (Table 3) than the binding energy of the substrate p-nitrophenol, underlining the strong inhibitory effects of these monoesters against SULT1A1*2. 

The distinct interactions of the DiNP metabolites with SULT1A1*2 compared to SULT1A1*1 may be due to the decreased flexibility of the active site of SULT1A1*2 [34], with MCiOP and MhiNP being higher than the other two metabolites. This outcome is also in good correlation with literature data revealing that an R213H mutation corresponding to SULT1A1*2 induces a local conformational change and results in lower catalytic activity of the enzyme [35].

In the case of SULT1A1*3, all DiNP metabolites bind to the active site of the enzyme. It was already shown that an M223V mutation corresponding to the SULT1A1*3 allelic variant produced an increase in the local flexibility of the active site of this enzyme [35]. This increased flexibility may add to the ability of this variant to accommodate all the DiNP metabolites, when compared to the SULT1A1*1 and SULT1A1*2 variants. This outcome is also in good agreement with information resulting from a molecular dynamics study and emphasizing that an M223V mutation resulted in the loss of a hydrophobic contact between M223 and M60 and led to increased local flexibility and the ability to accommodate larger compounds [5].

DiNP metabolites do not bind to the active sites of SULT1A3 and SULT1B1, respectively. As substrates, SULT1A3 has monoamines and biogenic catecholamines [36] that are hydrophilic compounds. Consequently, it is not an unexpected result that the hydrophobic DiNP metabolites are not able to bind to the active site of SULT1A3. The output of the molecular docking study regarding the binding of dopamine (the known substrate of SULT1A3) to SULT1A3 revealed that the binding mode obtained by docking corresponds to the position of dopamine in the crystallographic structure (Appendix A), with the binding energy being −9.41 kcal/mol and underlying the correct predictions obtained through docking. 

In the case of SULT1B1, the crystal structure that was used for molecular docking contains the enzyme in the complex with A3P and the substrate resveratrol. Table 1 and Figure 1 reveal dissimilar molecular properties and molecular lipophilic potential for resveratrol, compared to those of the DiNP metabolites. Consequently, the physicochemical properties of DiNP metabolites do not fit those of the active site of the enzyme. Another molecular docking study revealed that several other monoester phthalates (mono-hexyl phthalate, mono-octyl phthalate, mono-ethylhexyl phthalate, mono-butyl phthalate, mono-cyclohexyl phthalate) were able to inhibit SULT1B1 [16]. Furthermore, it was shown that phthalate monoesters, having shorter chains, significantly inhibited SULT1B1 but not the other SULT1 enzymes [16]. The compounds analyzed in the study of Huang et al. (2022) [16] are usually smaller than the DiNP metabolites and their physicochemical properties are more similar to those of resveratrol (Appendix A). The output of the molecular docking study regarding the binding of resveratrol (the known substrate of SULT1B1) to the SULT1B1 structure highlighted that the binding mode obtained by docking matches the position of resveratrol in the crystallographic structure (Appendix A), with the binding energy being −7.16 kcal/mol. It underlines the correct predictions obtained through docking.

All DiNP metabolites bind to the active sites of SULT1C1 and of SULT1E1, respectively. The binding of DiNP metabolites to SULT1C1 can be explained by the resemblance of their properties with those of the known SULT1C1 substrates: thyroid hormones and phenolic drugs [37]. The thyroid hormones are hydrophobic compounds and have a higher molecular weight (and consequently dimensions) compared to DiNP metabolites. In the case of SULT1E1, the crystal structure that was used in the molecular docking study contains the inhibitor 3,5,3′,5′-tetrachloro-biphenyl-4,4′-diol, a molecule that has similar physicochemical and geometric properties to those of the DiNP metabolites (see Table 1 and Figure 1). Moreover, the output of the molecular docking study regarding the binding of the inhibitor 3,5,3′,5′-tetrachloro-biphenyl-4,4′-diol to the SULT1E1 structure emphasized that the binding mode obtained by docking matches the position of the 3,5,3′,5′-tetrachloro-biphenyl-4,4′-diol in the crystallographic structure (Appendix A) and underlies the correct predictions obtained through docking. The binding energy of the inhibitor 3,5,3′,5′-tetrachloro-biphenyl-4,4′-diol to SULT1E1 is −6.91 kcal/mol, lower than the binding energies of the DiNP metabolites to the active site of this enzyme (Table 3). It sustains the inhibitory potential of DiNP metabolites against SULT1E1. Another study revealed that mono-hexyl phthalate, mono-octyl phthalate and mono-ethylhexyl phthalate were potent inhibitors of the activity of SULT1E1 [16].

SULT1A1*1, SULT1A1*2, SULT1A1*3, SULT1A2, SULT1C1 and SULT1E1 were inhibited by some or all of the DiNP metabolites. MiNP is the metabolite that inhibits almost all of these members of the SULT1 family, with the highest binding energy corresponding to SULT1A1*3. The stronger interaction of MiNp with SULT1A1*3 may be explained by the fact that, among the DiNP metabolites, MiNP has the lowest molecular weight and the highest hydrophobicity. It was revealed that anM223V mutation corresponding to the SULT1A1*3 allelic variant produced an increase in the local hydrophobicity of the catalytic site [34]. 

Data obtained by using the PLIP 2021 software emphasize the amino acids that are involved in the noncovalent interactions of the DiNP metabolites and SULT1 enzymes. This outcome allows a full description of the inhibition of SULT1 by these compounds. Taking into account the hydrophobic character of the DiNP metabolites, the predominance of hydrophobic contacts was expected.

The inhibition of phase I or/and phase 2 of metabolism may result in toxic concentrations of both endo- and xenobiotics. Furthermore, for numerous xenobiotics, the enzymes responsible for metabolizing them or/and the molecular targets of xenobiotics in humans are unknown. Predicting the molecular targets and what enzymes are required for metabolism allows for a better understanding of how humans handle a xenobiotic based on a given exposure.

## 4. Materials and Method

### 4.1. Materials

Several metabolites of di-isononyl phthalate (IUPAC name bis(7-methyloctyl) benzene-1,2-dicarboxylate) are considered in the present study (Figure 8): mono-carboxy-isooctyl phthalate (MCiOP, IUPAC name 2-(8-carboxyoctan-2-yloxycarbonyl)benzoic acid), mono-hydroxy-isononyl phthalate (MHiNP, IUPAC name 2-(1-hydroxy-7-methyloctoxy)carbonylbenzoate), mono-isononyl phthalate (MiNP, IUPAC name 2-(7-methyloctoxycarbonyl)benzoic acid) and mono-oxo-isononyl phthalate (MOiNP, IUPAC name 2-{[(4-methyl-7-oxooctyl)oxy]carbonyl}benzoic acid). Their structures have been extracted from PubChem database [38] and visualized using Chimera 1.16 software [39]. In order to implement the molecular docking study (see further), structures of SULT1 enzymes and of their allelic variants in complex with cofactor and ligands (if available) are also needed. These structures have been extracted from Protein Data Bank (PDB, [40]) and are presented in Table 5. All structures have been prepared for molecular docking using UCSF Chimera 1.16 software.

### 4.2. Extraction of the Physicochemical and Computation of the Structural Properties of the Metabolites of Di-Isononyl Phthalate and of the Ligands That Are Present in the Crystallographic Structures of SULT1 Enzymes

Physicochemical properties and three-dimensional structures of the metabolites of DiNP have been extracted from PubChem database [38]. Surface area and volume of these compounds have been computed using Chimera 1.16 software and their molecular lipophilicity potential has been obtained using Molinspiration Galaxy 3D Structure Generator v2021.01 (https://www.molinspiration.com/cgi/galaxy, accessed on 4 June 2023). For comparison purposes, the physicochemical and structural properties of ligands that are present in the crystallographic structures of SULT1 enzymes have been also obtained. 

### 4.3. Prediction of the Toxicological Effects of Di-Isononyl Phthalate and of Its Metabolites

ADMETLab2.0 server has been used in order to obtain information regarding the possible human organ toxicological effects of DiNP and of its metabolites. The classification models used for predicting the toxicity endpoints have a minimum accuracy of 80%; regarding the regression models, most of them achieved an R^2^ > 0.72. There are numerous other computational tools that can be used to assess the ADMET properties, with ADMETLab2.0 tool being used in this study because it is based on well-performed prediction models and has been used by more than 50,000 users in recent years, with millions of compounds screened [23,24]. The SMILES (Simplified Molecular Input Line Entry System) formulas of the metabolites of DiNP were used as the entry data, and the pharmacokinetics and toxicological endpoints under investigation were: clearance, penetration of the blood–brain barrier, possibility to be substrate or inhibitors of P-glycoprotein, plasma protein binding, hepatotoxicity, nephrotoxicity, cardiotoxicity, mutagenicity, carcinogenicity and endocrine effects. 

### 4.4. Evaluation of the Interactions of DiNP Metabolites with the Family 1 of Human Sulfotransferases

The molecular docking approach has been considered for assessing the possible interactions of the DiNP metabolites with the SULT1 family. In order to implement the molecular docking study, the structures of enzymes and those of the metabolites of di-isononyl phthalates are necessary. Structures of DiNP metabolites were extracted from PubChem database. Structures of SULT1 enzymes have been extracted from Protein Data Bank and are described in Table 5. All structures have been prepared for molecular docking using UCSF Chimera 1.16 software [38]. Molecular docking has been implemented using SwissDock web service [48] that is based on EADock algorithm [49]. An accurate and blind docking has been selected. 

It is known that both SULT1A1 and SULT1A2 have p-nitrophenol as a specific substrate; in order to interpret the docking outputs, structures of the SULT1A2 and of the variants SULT1A1*1 and SULT1A1*3 (that do not have a ligand in the structural file) have been superimposed to the structure of SULT1A1*2 that corresponds to the complex formed by the SULT1A1*2 variant with the inactive cofactor adenosine-3′-5′-diphosphate (A3P) and two molecules of p-nitrophenol (Appendix A). The superimposition reveals a high similarity between the structures and emphasized the most probable position of p-nitrophenol when binding to SULT1A1*1, SULT1A1*3 and SULT1A2, respectively. Consequently, the binding modes for the DiNP metabolites that resulted from molecular docking were compared to the position occupied by the two p-nitrophenol molecules that are present in the crystallographic structure of their complex with SULT1A1*2. 

In the case of SULT1C1, in order to locate the catalytic cavity and to more easily interpret the molecular docking outcomes, its crystal structure has been superposed with both that of SULT1A1*2 containing the inactive cofactor A3P and two molecules of p-nitrophenol, and that of SULT1B1 that contains the substrate resveratrol, with the superposition also reflecting a high similarity between the three structures (Appendix A). Furthermore, as a control, molecular docking has been implemented for every enzyme with its substrate/inhibitor that is present in the crystallographic structure, when available. In the case of SULT1C1 as a control, molecular docking has been implemented for assessing its interaction with its known substrate, iodothyronine [37]. 

The noncovalent interactions between the DiNP metabolites and the SULT1 enzymes have been analyzed using the Protein–Ligand Interaction Profiler tool [50].

## 5. Conclusions

The results obtained in the present study reveal several human health risks of the metabolites of di-isononyl phthalate. These compounds revealed the capacity to cause liver damage, to inhibit receptors activated by peroxisome proliferators and are usually able to inhibit the activity of family 1 of sulfotransferases, except SULT1A3 and SULT1B1. Taking into account that humans are extensively exposed to phthalates and the important role of sulfotransferases for the detoxification and elimination of numerous endogenous and exogenous molecules in the body, information regarding the biological effects of the metabolites of phthalates and their possible inhibitory effects against the activities of sulfotransferases contributes to better understanding the risk of human exposure to phthalates. 

## Figures and Tables

**Figure 1 molecules-28-06748-f001:**
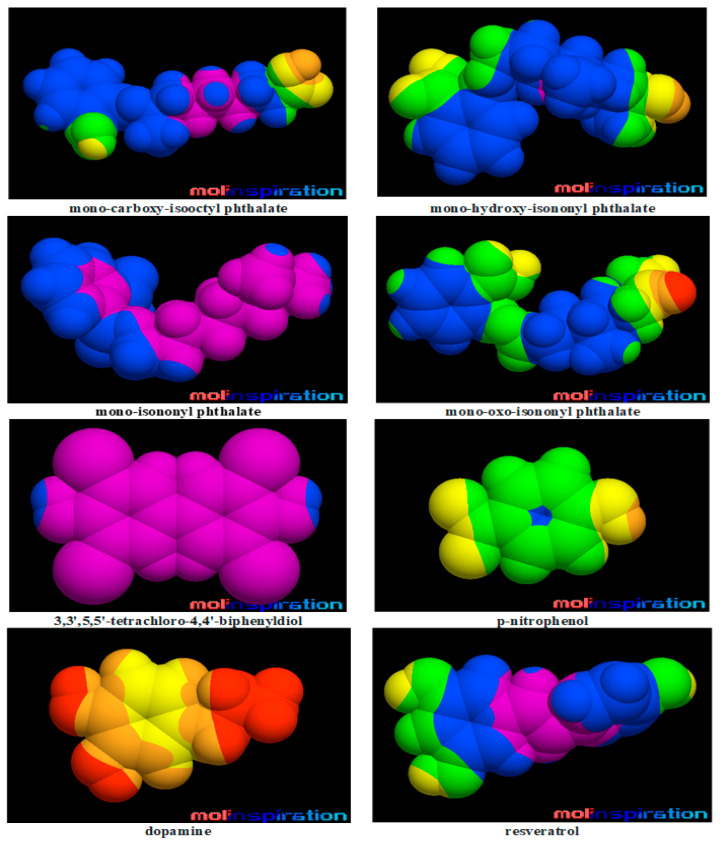
Molecular lipophilicity potential on the molecular surface for the metabolites of di-isononyl phthalate and for the ligands that are present in the crystallographic structures of SULT1 enzymes. The hydrophobic parts of the surface are colored by violet and blue and the hydrophilic ones by orange and red.

**Figure 2 molecules-28-06748-f002:**
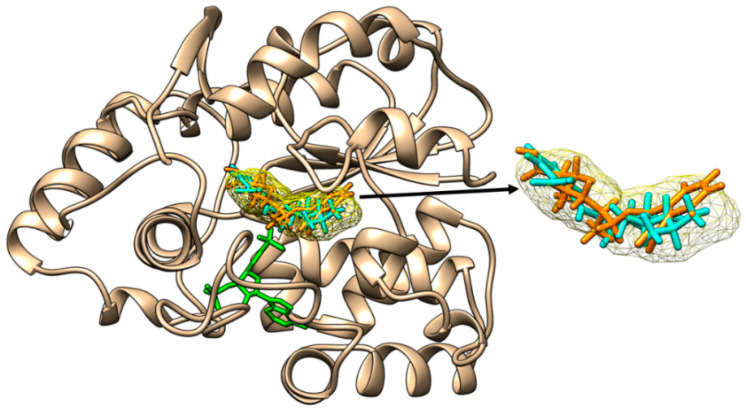
Binding of mono-carboxy-isooctyl phthalate (orange sticks) and of mono-isononyl phthalate (cyan sticks) to the active site of SULT1A1*1. The binding poses correspond to the binding position of p-nitrophenol (yellow mesh surface) to SULT1A1*2 and superimposed on the structure of SULT1A1*1. The inactive cofactor adenosine-3′-5′-diphosphate that is present in the structural file of SULT1A1*1 is revealed in green sticks.

**Figure 3 molecules-28-06748-f003:**
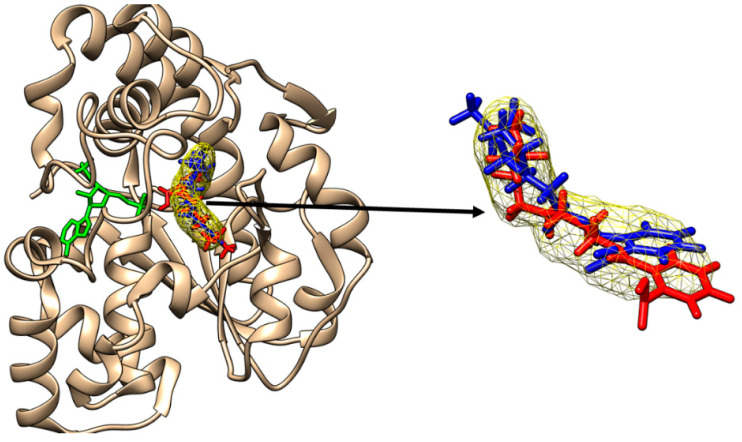
Binding of mono-isononyl phthalate (red sticks) and of mono-oxo-isononyl phthalate (blue sticks) to the active site of SULT1A1*2. The binding poses correspond to the binding position of p-nitrophenol (yellow mesh surface). The inactive cofactor adenosine-3′-5′-diphosphate is revealed in green sticks.

**Figure 4 molecules-28-06748-f004:**
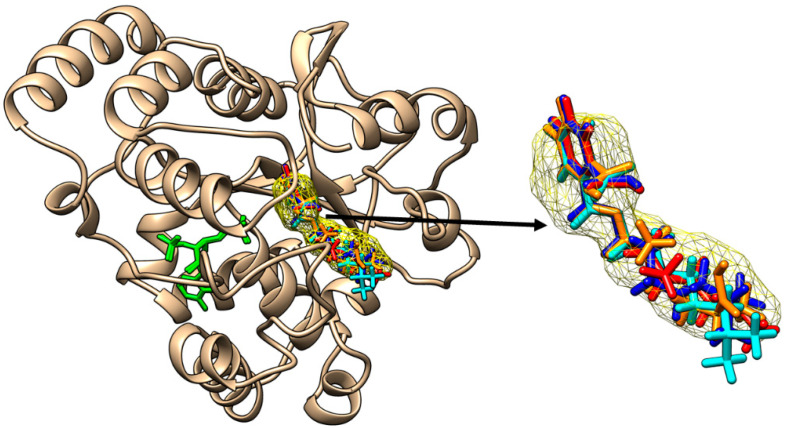
Binding of mono-carboxy-isononyl phthalate (orange sticks), of mono-hidroxy-isononyl phthalate (cyan sticks), of mono-isononyl phthalate (blue sticks) and of mono-oxo-isononyl phthalate (red sticks) to the active site of SULT1A1*3. All the binding poses correspond to the binding position of p-nitrophenol (yellow mesh surface). The inactive cofactor adenosine-3′-5′-diphosphate is revealed in green sticks.

**Figure 5 molecules-28-06748-f005:**
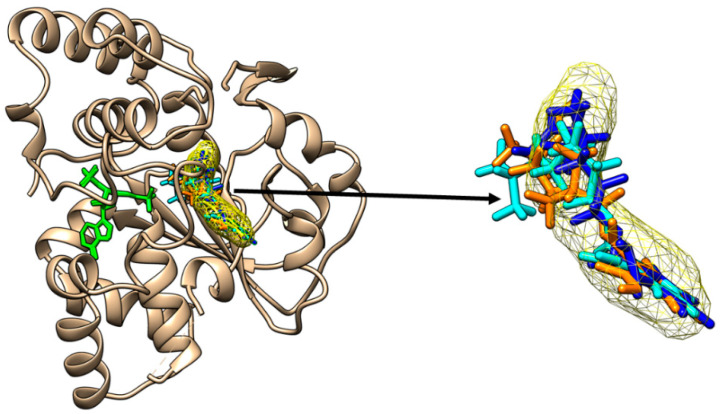
Binding of mono-carboxy-isononyl phthalate (orange sticks), of mono-isononyl phthalate (cyan sticks) and of mono-oxo-isononyl phthalate (blue sticks) to the active site of SULT1A2. All the binding poses correspond to the binding position of p-nitrophenol (yellow mesh surface) to SULT1A1*2. The inactive cofactor adenosine-3′-5′-diphosphate is revealed in green sticks.

**Figure 6 molecules-28-06748-f006:**
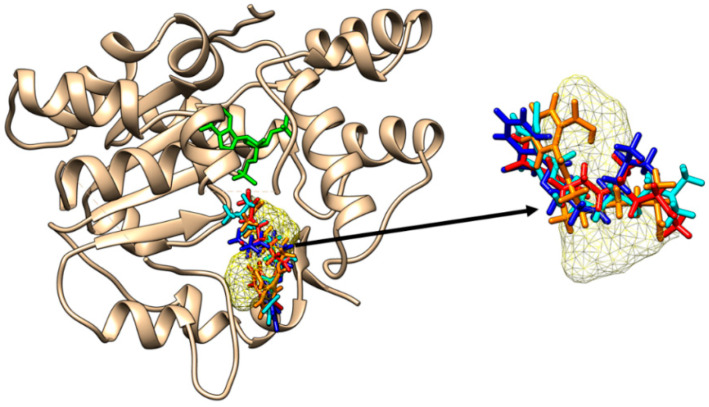
Binding of mono-carboxy-isononyl phthalate (orange sticks), of mono-hidroxy-isononyl phthalate (cyan sticks), of mono-isononyl phthalate (blue sticks) and of mono-oxo-isononyl phthalate (red sticks) to the active site of SULT1C1. All the binding poses correspond to the binding position of iodothyronine (yellow mesh surface). The inactive cofactor adenosine-3′-5′-diphosphate is revealed in green sticks.

**Figure 7 molecules-28-06748-f007:**
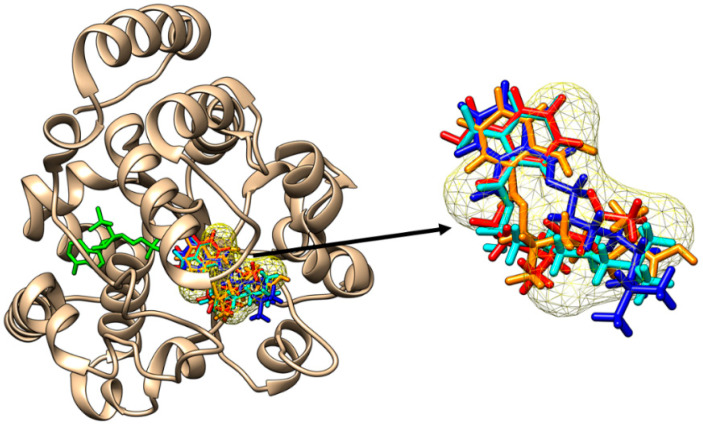
Binding of mono-carboxy-isononyl phthalate (orange sticks), of mono-hidroxy-isononyl phthalate (cyan sticks), of mono-isononyl phthalate (blue sticks) and of mono-oxo-isononyl phthalate (red sticks) to the active site of SULT1A1E1. All the binding poses correspond to the binding position of 3,3′,5,5′-tetrachloro-4,4′-biphenyldiol (yellow mesh surface). The inactive cofactor adenosine-3′-5′-diphosphate is revealed in green sticks.

**Figure 8 molecules-28-06748-f008:**
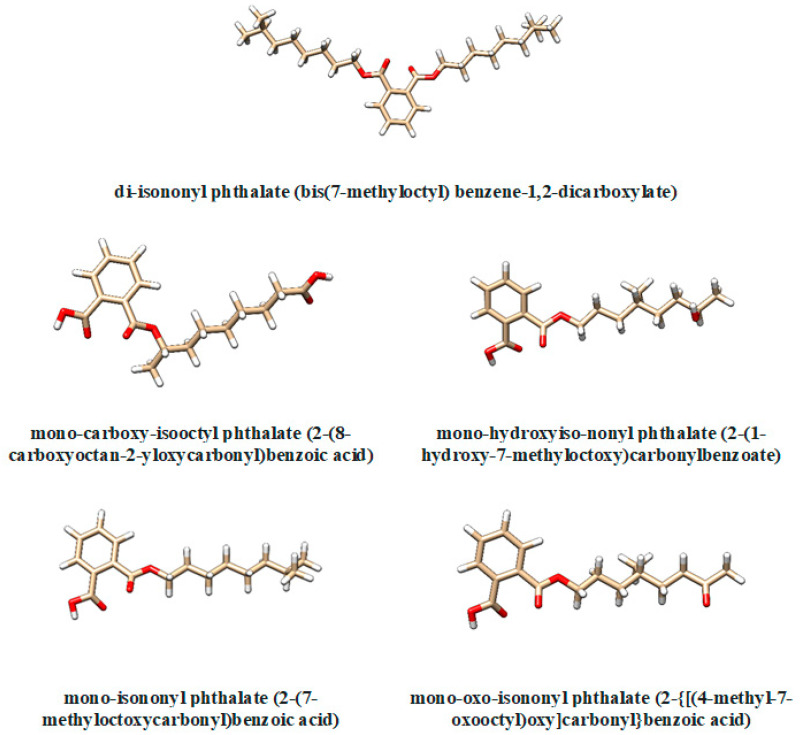
Three-dimensional structures of the di-isononyl phthalate and of its metabolites. The atoms are coloured like this: carbon—brown, oxygen—red, hydrogen—white.

**Table 1 molecules-28-06748-t001:** Physicochemical and structural properties of the metabolites of di-isononyl phthalate and of the ligands that are present in the crystallographic structures of SULT1 enzymes: MW—molecular weight, logP—partition coefficient, HBD—hydrogen bonds donors, HBA—hydrogen bonds acceptors, RB—rotatable bonds, tPSA—topological polar surface area.

Compound	MW (g/mol)	LogP	HBD	HBA	RB	tPSA (Å^2^)	Area (Å^2^)	Volume (Å^3^)
mono-carboxy-isooctyl phthalate	322.40	3.4	2	6	10	101.0	285.1	287.3
mono-hydroxy-isononyl phthalate	308.40	3.9	2	5	10	83.8	293.5	284.6
mono-isononyl phthalate	292.40	5.6	1	4	10	63.6	288.8	268.6
mono-oxo-isononyl phthalate	306.40	2.8	1	5	10	80.7	278.8	288.0
3,3′,5,5′-tetrachloro-4,4′-biphenyldiol	324.00	5.4	2	2	1	40.5	265.5	252.3
p-nitrophenol	139.11	1.9	1	3	0	66.0	128.6	106.1
dopamine	153.18	−1.0	3	3	2	66.5	156.8	132.4
resveratrol	228.24	3.1	3	3	2	60.7	217.7	191.0

**Table 2 molecules-28-06748-t002:** Probabilities regarding the possible toxicological effects of DiNP metabolites: CL—clearance, BBB—blood–brain barrier penetration, PPB—plasma protein binding, PgpI—inhibitor of P-glycoprotein, PgpS—substrate of P-glycoproteine, hERG—human cardiotoxicity, DILI—induced liver damage, NR-AR-LBD—binding domain of androgen receptor ligands, NR-AhR—aryl hydrocarbon receptor, NR-aromatase—endocrine disrupting chemicals, NR-ER-LBD—estrogen receptor ligand binding domain, NR-PPAR-gamma—activated receptors of peroxisome proliferators. Cells colored in orange reveal mean toxicological effects.

Compound/Toxicity	Di- Isononyl Phthalate	Mono-Carboxy- Isooctyl Phthalate	Mono-Hydroxy- Isononyl Phthalate	Mono- Isononyl Phthalate	Mono-Oxo -Isononyl phthalate
CL (mL/min)	8.329	1.369	7.210	3.032	5.149
BBB	0.011	0.109	0.379	0.106	0.336
PPB (%)	97.84	95.97	90.69	97.34	92.87
PgpI	0.974	0.004	0.003	0.006	0.026
PgpS	0.000	0.001	0.026	0.000	0.004
hERG	0.190	0.053	0.209	0.260	0.198
DILI	0.358	0.795	0.743	0.852	0.770
Ames mutagenicity	0.002	0.007	0.004	0.004	0.005
Carcinogenicity	0.302	0.019	0.019	0.300	0.017
NR-AR-LBD	0.002	0.060	0.007	0.003	0.007
NR-AhR	0.071	0.012	0.015	0.057	0.042
NR-aromatase	0.086	0.005	0.009	0.011	0.008
NR-ER-LBD	0.230	0.064	0.036	0.010	0.012
NR-PPAR-gamma	0.060	0.744	0.890	0.872	0.785

**Table 3 molecules-28-06748-t003:** Binding energies of DiNP metabolites to the active sites of SULT1 enzymes.

Enzyme	ΔG (kcal/mol)			
	Mono-Carboxy-Isooctyl Phthalate	Mono-Hydroxy-Isononyl Phthalate	Mono-Isononyl Phthalate	Mono-Oxo-Isononyl Phthalate
SULT1A1*1	−8.06	-	−6.79	-
SULT1A1*2	-	-	−7.51	−8.30
SULT1A1*3	−8.36	−11.36	−8.46	−8.29
SULT1A2	−7.3	-	−7.75	−6.98
SULT1C1	−8.04	−9.99	−7.60	−7.76
SULT1E1	−7.95	−10.91	−7.54	−7.83

**Table 4 molecules-28-06748-t004:** Amino acids belonging to SULT1 enzymes and involved in the noncovalent interactions with their substrates/inhibitors and the DiNP metabolites: MCiOP—monocarboxy-izooctyl phthalate, MHiNP—monohydroxy-isononyl phthalate, MiNP—mono-isononyl phthalate, MOiNP—monooxo-isononyl phthalate, PNP—p-nitrophenol, TBD—3,5,3′,5′-tetrachloro-biphenyl-4,4′-diol.

Complex	Hydrophobic Contacts	Hydrogen Bonds	π Staking	π Cations	Salt Bridges
SULT1A1*1—MCiOP	PHE76, PHE81, PHE84, ILE89, PHE142	TYR139, TYR140	PHE84	-	LYS106, HIS149
SULT1A1*1—MiNP	ILE21, PRO47, PHE76, PHE84, PHE142, TYR169, TYR240, VAL243, PHE214, PHE255	-	-	-	LYS106
SULT1A1*2—PNP in the crystallographic structure	PHE76, PHE84, ILE89, VAL243, PHE247	-	PHE76	-	-
SULT1A1*2—PNP as a result of docking	PHE76, PHE84, ILE89, VAL243, PHE247	-	PHE76	-	-
SULT1A1*2—MiNP	PRO47, PHE76, PHE81, PHE84, ILE89, PHE142, TYR240,	LYS48	PHE142	-	LYS48, LYS106
SULT1A1*2—MOiNP	ILE21, PHE24, PRO47, PHE76, PHE81, PHE84, ILE89, TYR139, ALA146, VAL148, TYR169	TYR240	PHE84	-	LYS106
SULT1A1*3—MCiOP	ILE21, PHE24, PRO47, PHE76, PHE84, ILE89, PHE142, TYR169, TYR240, VAL243	-	-	-	LYS106
SULT1A1*3—MHiNP	PRO47, PHE76, PHE84, ILE89, PHE142, VAL243, PHE247	TYR240	-	-	LYS48, LYS106 (3)
SULT1A1*3—MiNP	ILE21, PHE76, PHE81, PHE84, ILE89, PHE142, TYR169, TYR240, VAL243	-	-	-	LYS106
SULT1A1*3—MOiNP	ILE21, PHE24, PHE76, PHE81, PHE84, ILE89, VAL148, TYR149	-	PHE84	-	LYS106
SULT1A2—MCiOP	PHE76, PHE81, PHE84, ILE89, THR95	TYR240 (2)	-	-	LYS106
SULT1A2—MiNP	LYS48, PHE76, PHE81, ILE89, PHE142, PHE255	-	PHE84	-	LYS106
SULT1A2—MOiNP	ILE21, PHE24, PHE76, PHE81, PHE84, ILE89, VAL148, TYR149	-	PHE84	-	LYS106
SULT1C1—iodothyronine	PHE82, MET149, TRP 170	GLN22	-	-	LYS96
SULT1C1—MCiOP	TRP145, PHE143, LEU150	LYS49, ASN147	-	-	LYS49
SULT1C1—MHiNP	PHE143, MET149	PHE256	-	-	LYS49 (3)
SULT1C1—MiNP	GLN22, ALA24, ARG87, PHE143, TRP170	THR25, TRP85, ALA86	-	-	-
SULT1C1-MOiNP	GLN22, PHE82, PHE143, TRP170	GLN22, TRP85	-	-	-
SULT1E1—TBD in crystallographic structure	TYR20, PHE23, VAL145	LYS105, HIS107	PHE141	HIS148	-
SULT1E1—TBD as a result of docking	TYR20, ASP22, PHE23, PHE141, VAL145, ALA146, TYR239, ILE246	ASP22, LYS105, HIS107	PHE80, PHE141	HIS148	
SULT1E1—MCiOP	PHE23, PRO46, LYS47, LYS85PHE141, ILE246	TYR239	-	-	LYS105
SULT1E1—MHiNP	TYR20, PRO46, LYS47, PHE80, PHE141, VAL145	LYS47, TYR239	-	-	LYS47, LYS105 (3)
SULT1E1—MiNP	PHE23, PRO46, LYS47, PHE141, VAL175, PHE254	-	-	-	LYS105
SULT1E1—MOiNP	TYR20, PRO46, LYS47, PHE80, PHE141, VAL145, ILE246	LYS47, THR50	-	-	LYS47, LYS105 (2)

**Table 5 molecules-28-06748-t005:** Structural files available in Protein Data Bank (PDB) corresponding to SULT1 enzymes and their allelic variants in complex with cofactor and substrates/ligands. PAP—the inactive cofactor adenosine-3′-5′-diphosphate.

PDB ID of the Structural File	Description	References
4GRA	Crystal structure of SULT1A1*1 (wild type) complexed with PAP	[41]
1LS6	Crystal structure of SULT1A1*2 complexed with PAP and the substrate p-pitrophenol	[42]
1Z28	Crystal structure of SULT1A1*3 complexed with PAP	[43]
1Z29	Crystal structure of SULT1A2 complexed with PAP, acetic acid and Ca^2+^	
2A3R	Crystal structure of SULT1A3 in complex with PAP and the substrate dopamine	[44]
3CKL	Crystal structure SULT1B1 in complex with PAP and the substrate resveratrol	[45]
3BFX	Crystal structure SULT1C1 in complex with PAP	[46]
1G3M	Crystal structure SULT1E1 in complex with PAP and the inhibitor 3,5,3′,5′-tetrachloro-biphenyl-4,4′-diol	[47]

## Data Availability

All the data are available in the article and the Appendix A.

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
