# Peer review of "Evaluation of the Toxicity Potential of the Metabolites of Di-Isononyl Phthalate and of Their Interactions with Members of Family 1 of Sulfotransferases—A Computational Study"

_molecules, 2023, doi:10.3390/molecules28186748_

Round 1
Reviewer 1 Report
The study evaluates the toxicity potential of metabolites of di-isononyl phthalate (DiNP) and their interactions with human sulfotransferase enzymes (SULTs).
Four DiNP metabolites were considered: mono-carboxy-isooctyl phthalate (MCiOP), mono-hydroxy-isononyl phthalate (MHiNP), mono-isononyl phthalate (MiNP), and mono-oxo-isononyl phthalate (MOiNP). Computer modeling was used to predict the toxicity of the metabolites. All four showed the potential to cause liver damage and inhibit receptors activated by peroxisome proliferators. Molecular docking assessed the inhibition of SULT enzymes. The metabolites could inhibit SULT1A1, SULT1A2, SULT1A3, SULT1C1 and SULT1E1 but not SULT1A3 and SULT1B1.
The strongest binding was seen between MiNP and SULT1A3. The metabolites were predicted to bind similarly to a known SULT1E1 inhibitor.
Overall, the manuscript contains a large body of work using computer simulation. thereof, and a comprehensive analysis of computer simulation. However, I have several major concerns, as outlined below:
Major comments:
1. Throughout the full text articles all chapters are based on computer simulation and indicators of humans. I think we can compete Validation in human cell (Such as HaCat) Complete the indicators, such as Western-Blotting IF-staining or RT-PCR, will make our results more competitive and more persuasive. With downstream function verification, I believe that the molecules of the experiment become vivid rise. SULT1A1 site, for example, can be in the COSMIC web site at https://cancer.sanger.ac.uk/cosmic to find the sequence, then build the mutations of plasmid, after a series of functional verification for cells. This is a very quick experiment.
2.I think why not try to simulate the drug in mice? such as HE dyeing Western-Blotting IF-staining or RT-PCR, This is a very quick experiment, only one may be solve a lot of problem.
All in all, Verification without biology is bound to raise questions from the reviewer.
Minor comments:
42- focusses → focuses
70-may lead to harm effects more better than word conduct.
207 lead→ led
Author Response
The study evaluates the toxicity potential of metabolites of di-isononyl phthalate (DiNP) and their interactions with human sulfotransferase enzymes (SULTs).
Four DiNP metabolites were considered: mono-carboxy-isooctyl phthalate (MCiOP), mono-hydroxy-isononyl phthalate (MHiNP), mono-isononyl phthalate (MiNP), and mono-oxo-isononyl phthalate (MOiNP). Computer modeling was used to predict the toxicity of the metabolites. All four showed the potential to cause liver damage and inhibit receptors activated by peroxisome proliferators. Molecular docking assessed the inhibition of SULT enzymes. The metabolites could inhibit SULT1A1, SULT1A2, SULT1A3, SULT1C1 and SULT1E1 but not SULT1A3 and SULT1B1.
The strongest binding was seen between MiNP and SULT1A3. The metabolites were predicted to bind similarly to a known SULT1E1 inhibitor.
Overall, the manuscript contains a large body of work using computer simulation. thereof, and a comprehensive analysis of computer simulation. However, I have several major concerns, as outlined below.
Thank you very much for the suggestions and observations that are clearly meant to improve the quality of the manuscript. We answer to them using text in blue in the revised manuscript.
Major comments:
- Throughout the full text articles all chapters are based on computer simulation and indicators of humans. I think we can compete Validation in human cell (Such as HaCat) Complete the indicators, such as Western-Blotting IF-staining or RT-PCR, will make our results more competitive and more persuasive. With downstream function verification, I believe that the molecules of the experiment become vivid rise. SULT1A1 site, for example, can be in the COSMIC web site at https://cancer.sanger.ac.uk/cosmic to find the sequence, then build the mutations of plasmid, after a series of functional verification for cells. This is a very quick experiment.
2.I think why not try to simulate the drug in mice? such as HE dyeing Western-Blotting IF-staining or RT-PCR, This is a very quick experiment, only one may be solve a lot of problem.
All in all, Verification without biology is bound to raise questions from the reviewer.
The Referee is right, verification with biology strengthen the validity of the outputs of computational studies. Unfortunately, we are not able to perform any of the mentioned test, these techniques are not available in our institution. In order to be clear for readers that the manuscript reports results obtained by strictly computational methods, the title of the manuscript has been changed in “Evaluation of the toxicity potential of the metabolites of di-isononyl phthalate and of their interactions with members of family 1 of sulfotransferases. A computational study”. Computational methods are considered by regulatory agencies as being capable of predicting toxicity, identifying, assessing, and managing risk of chemicals, Consequently, the Organization of Economic and Co-operation Development (OECD) created quantitative structure-activity relationship (QSAR) guidelines already in 2004 and the principles for the construction of (Q)SAR models, computational methods their self, and model validation methods are described in detail since 2007. Furthermore, REACH (Registration, Evaluation and Authorization of CHemicals) regulation mentions the computational techniques for studying the toxicological profile of chemicals. Not at last, molecular modeling methods offer the possibility to evaluate the risk to human health due to the exposure to various chemicals as they can simulate possible modes of action. All these virtual screening tools can help prioritize bioassay requirements.
Minor comments:
42- focusses → focuses
70-may lead to harm effects more better than word conduct.
207 lead→ led
All these corrections are implemented in the revised manuscript.
Reviewer 2 Report
Humans are exposed to the environment's potentially toxic chemicals, di-isononyl phthalates. Sulfotransferases (SULT) are enzymes responsible for the detoxification of these compounds. Then, the authors evaluate the toxicity of di-isononyl phthalate metabolites as well as the interaction with SULTs by the computational approach. Di-isononyl phthalate's metabolites may cause liver damage and inhibit receptors activated by peroxisome. These theoretical model approaches could be important for evaluating these potentially toxic chemicals.
Points to be addressed
1. Lines 131-141: The prediction of toxicity seems to be made by ADMETLab2.0 server. Please explain how ADMETLab2.0 works and why the authors chose this system.
2. Table 2: Please include which system was used for the prediction in the Table's caption. It is not clear what exactly each figure of probability means. For instance, what does ‘0.974’ of di-isononyl Phthalate in PgpI suggest? Probability to what? In addition, What does NR stand for in NR-AR-LBD. It is probably for nuclear receptors. But, then, what is NR-aromatase?
3. Table 4: As the same as above, please explain which systems were used for the calculation in Table’s caption.
4. Please correct the numbering of Tables and Figures.
5. Maybe the Discussion section could be separated from the results section. Because plenty of results are presented, it would be easier for readers to understand the authors’s line of thought easily with a separate Discussion section.
English needs to be improved. Even Abstract is not easy to read through.
Author Response
Comments and Suggestions for Authors
Humans are exposed to the environment's potentially toxic chemicals, di-isononyl phthalates. Sulfotransferases (SULT) are enzymes responsible for the detoxification of these compounds. Then, the authors evaluate the toxicity of di-isononyl phthalate metabolites as well as the interaction with SULTs by the computational approach. Di-isononyl phthalate's metabolites may cause liver damage and inhibit receptors activated by peroxisome. These theoretical model approaches could be important for evaluating these potentially toxic chemicals.
Thank you very much for the suggestions and observations that are clearly meant to improve the quality of the manuscript. We answer to them using text in green in the revised manuscript. We hope that the reviewer’s suggestions have been correctly understood.
Points to be addressed
- Lines 131-141: The prediction of toxicity seems to be made by ADMETLab2.0 server. Please explain how ADMETLab2.0 works and why the authors chose this system.
The information has been added (lines 134-142) for explaining how this computational tool works. The argument for using this tool has been added in the Methodology section, subsection 3.3, lines 413-419.
- Table 2: Please include which system was used for the prediction in the Table's caption. It is not clear what exactly each figure of probability means. For instance, what does ‘0.974’ of di-isononyl Phthalate in PgpI suggest? Probability to what? In addition, What does NR stand for in NR-AR-LBD. It is probably for nuclear receptors. But, then, what is NR-aromatase?
The requested information has been added in the text, lines 143-154 and 163-172.
- Table 4: As the same as above, please explain which systems were used for the calculation in Table’s caption.
The requested information has been added in the text, lines 244-245
- Please correct the numbering of Tables and Figures.
The tables and figures are renumbered in the entire manuscript and in the Supplementary material.
- Maybe the Discussion section could be separated from the results section. Because plenty of results are presented, it would be easier for readers to understand the authors’s line of thought easily with a separate Discussion section.
The Results and Discussions section has been divided into separate Results and Discussions sections, as suggested.
Comments on the Quality of English Language
English needs to be improved. Even Abstract is not easy to read through.
English has been revised, we did our best to correct the text. The sentences have been shortened and/or reformulated. All the modified/added text is in green.
Round 2
Reviewer 1 Report
- I still insist on biological experiments, but I think it's okay to publish them this way. I generally agree with publication.
Author Response
Thank you again for your suggestion, we would have been happy to implement it. As I have already explained, unfortunately we are not able to perform the suggested experiments. Furthermore, there is not possible to perform these experiments in the labs of our local research partners. We take into account to enrich our infrastructure and team members in the next years such as to be able to implement experimental studies, but the actual PhD student should finish her thesis and cannot wait so long. We did a lot of work to obtain these results, and there are numerous computational studies that are not accompanied by experiments and are still considered for publication in respected journals. In our opinion, this is not a reason why our manuscript should not be accepted. Last but not least, our manuscript fits the purpose of the special issue, Advances in Molecular Modeling in Chemistry, for which we applied.
Within the revised mansucript, there is a new paragraph explaining the reliability of the computational methods, as the Editor asked. The text is in orange (lines 102-120).